# Hemoadsorption Therapy for Critically Ill Patients with Acute Liver Dysfunction: A Meta-Analysis and Systematic Review

**DOI:** 10.3390/biomedicines12010067

**Published:** 2023-12-27

**Authors:** Caner Turan, Csenge Erzsébet Szigetváry, Tamás Kói, Marie Anne Engh, Işıl Atakan, László Zubek, Tamás Terebessy, Péter Hegyi, Zsolt Molnár

**Affiliations:** 1Department of Anesthesiology and Intensive Therapy, Semmelweis University, 1085 Budapest, Hungary; c.caner.turan@gmail.com (C.T.); szigetvary.csenge@gmail.com (C.E.S.); 2Centre for Translational Medicine, Semmelweis University, 1085 Budapest, Hungary; samatiok@gmail.com (T.K.); islatakan@gmail.com (I.A.); laszlo.zubek@gmail.com (L.Z.); drterebessytamas@yahoo.com (T.T.); hegyi2009@gmail.com (P.H.); 3Department of Stochastics, Institute of Mathematics, Budapest University of Technology and Economics, 1111 Budapest, Hungary; 4Department of Orthopaedics, Semmelweis University, 1085 Budapest, Hungary; 5Institute of Pancreatic Diseases, Semmelweis University, 1085 Budapest, Hungary; 6Institute for Translational Medicine, Medical School, University of Pécs, 7623 Pécs, Hungary; 7Department of Anesthesiology and Intensive Therapy, Poznan University of Medical Sciences, 60-806 Poznan, Poland

**Keywords:** hemoadsorption, liver dysfunction, critical care, database meta-analysis

## Abstract

Critically ill patients are at risk of developing acute liver dysfunction as part of multiorgan failure sequelae. Clearing the blood from toxic liver-related metabolites and cytokines could prevent further organ damage. Despite the increasing use of hemoadsorption for this purpose, evidence of its efficacy is lacking. Therefore, we conducted this systematic review and meta-analysis to assess the evidence on clinical outcomes following hemoadsorption therapy. A systematic search conducted in six electronic databases (PROSPERO registration: CRD42022286213) yielded 30 eligible publications between 2011 and 2023, reporting the use of hemoadsorption for a total of 335 patients presenting with liver dysfunction related to acute critical illness. Of those, 26 are case presentations (*n* = 84), 3 are observational studies (*n* = 142), and 1 is a registry analysis (*n* = 109). Analysis of data from individual cases showed a significant reduction in levels of aspartate transaminase (*p* = 0.03) and vasopressor need (*p* = 0.03) and a tendency to lower levels of total bilirubin, alanine transaminase, C-reactive protein, and creatinine. Pooled data showed a significant reduction in total bilirubin (mean difference of −4.79 mg/dL (95% CI: −6.25; −3.33), *p* = 0.002). The use of hemoadsorption for critically ill patients with acute liver dysfunction or failure seems to be safe and yields a trend towards improved liver function after therapy, but more high-quality evidence is crucially needed.

## 1. Introduction

Critically ill patients admitted to the intensive care unit (ICU) have been shown to be at risk of developing acute liver dysfunction usually as part of multiorgan failure sequelae [1]. Affecting at least 20% of patients, ICU-acquired liver dysfunction therefore has a frequent occurrence in the critically ill population and represents a life-threatening condition associated with a significantly increased risk of death [2,3]. In fact, early liver dysfunction, even after correction for other organ failures, is responsible for a mortality of 11% [4].

During such hyperinflammatory conditions, the liver is both a site of production of pro-inflammatory cytokines (TNF-α, IL-1β, IL-6) and a target organ for the effects of inflammatory mediators derived from extrahepatic sources of infection [5]. When advancing into more severe states, liver dysfunction can lead to hepatic encephalopathy or brain dysfunction as an expression of acute liver failure [6]. Furthermore, the disruption of the balance of reductive oxygen species is found to be implicated in biochemical and biophysical changes that might play a role in the progression of liver dysfunction into such severe disease states [7,8]. 

Bedside monitoring of the liver function of critically ill patients is not easy. Bilirubin represents the standard measure for the assessment of liver dysfunction in the ICU and is routinely assessed as part of the sequential organ failure assessment (SOFA) score since increased bilirubin plasma levels reflect a derangement in metabolic processes such as bile formation, bile secretion, and reduced bile flow into the biliary tract, the latter being considered the main component of early hepatic dysfunction under hyperinflammatory conditions [9,10]. However, despite good correlations between bilirubin plasma concentrations and mortality in several critically ill conditions (0.1–0.4 mg/dL total bilirubin was associated with higher cancer mortality (HR, 1.94; *p* = 0.016), whereas ≥0.8 mg/dL was associated with non-cancer, non-cardiovascular mortality (HR, 1.88; *p* = 0.002)) [11], bilirubin is a lagging parameter as there is a significant time lag between imminent or even established liver dysfunction and development of hyperbilirubinemia [12]. Thus, given the lack of diagnostic accuracy of standard laboratory parameters, diagnosis and monitoring of liver dysfunction in critically ill patients remains a major challenge with a very inconsistent definition and lack of clear diagnostic criteria [13].

Up to now, there is no specific therapy for acute liver dysfunction in critically ill patients, integrated management strategies and therapeutic interventions are hardly supported by randomized studies, and treatment is often center-specific [14]. Current clinical practice therefore focuses on timely decisions around transplant in conjunction with optimal multiple organ supportive care and effective therapeutic interventions.

Hemoadsorption is a new extracorporeal blood purification modality. It has been primarily used for cytokine adsorption to control hyperinflammation [15,16,17]. Acquired acute liver dysfunction in critically ill patients is also thought to be due to hyperinflammation [18]. Therefore, theoretically, clearing the blood from toxic liver-related metabolites and cytokines could be beneficial in improving liver function in this patient population. However, evidence of its efficacy is lacking, and despite its increasing use and accumulating data, a comprehensive summary on hemoadsorption in this setting is missing. 

### Objectives

The aim of this systematic review and meta-analysis is to assess the effect of hemoadsorption on clinical outcomes and the removal of total bilirubin, as well as the reduction in liver transaminases in critical illness-associated acute liver dysfunction.

## 2. Methods

We report our systematic review and meta-analysis in accordance with the PRISMA 2020 Statement (Appendix A: PRISMA 2020 Checklist), and it was conducted following the recommendations of the Cochrane Handbook for Systematic Reviews of Interventions [19].

### 2.1. Search Strategy

Two systematic literature searches were conducted on 18 February 2022 and 24 February 2023, using the following databases: Medline (via PubMed), Embase, Scopus, CENTRAL, and Web of Science (PROSPERO registration: CRD42022286213). The following search key was used in these databases: oXiris OR Jafron OR CytoSorb OR hemadsorption OR hemoadsorption OR “blood purification” OR “cytokine removal” AND liver failure OR “liver injury” OR liver dysfunction OR “hepatocellular injury” OR hepatic insufficiency OR hepatic dysfunction OR “acquired liver injury”. 

CytoSorb Literature Database, and the references of included studies, citing articles, authors’ other accessible publications, and ResearchGate were hand-searched for further eligible publications. No filters or restrictions were imposed on the search. 

### 2.2. Eligibility Criteria

Primary research publications with original clinical data were eligible for inclusion in this systematic review. Publications without original clinical data, such as reviews, commentaries, editorials, consensus, and guidelines, were excluded. Inclusion and exclusion criteria were framed beforehand in the PICO model (patients; intervention; control; outcomes). The target population was adult patients with acute liver dysfunction or failure associated with critical illness and treated with hemoadsorption (HA). Selected articles had to report one or more of the following to assess the effect of HA therapy: requirements of vasopressors, serum levels of bilirubin, and the liver enzymes alanine aminotransferase (ALT) or aspartate aminotransferase (AST) for pre- and post-hemoadsorption treatment. Primary outcomes were the change in liver function parameters during HA and mortality. We pooled data from individual cases to assess the variations in vasopressor needs and serum levels of bilirubin, ALT, and AST, before and after treatment with hemoadsorption, without considering the heterogeneity existing among different sources. In addition to the case studies, a pooled analysis was conducted for studies including data on control cohorts. The effect size was expressed as the mean difference in the relative changes of the aforementioned variables from baseline to post-treatment values.

### 2.3. Selection Process

The selection was performed by two independent review authors (CT as review author 1 and CS as review author 2). The two reviewer groups then assessed the results for inclusion, first by title and abstract selection, followed by full-text selection using the EndNote 20 software (Clarivate Analytics, Philadelphia, PA, USA). Any disagreements were resolved firstly by consensus between the reviewers or by a third independent investigator (FD) when needed. To evaluate inter-reviewer agreement, Cohen’s Kappa was calculated with the result being κ = 0.89 after full-text selection. 

### 2.4. Data Collection Process

From the eligible articles, data were collected by the two review authors (CT and IA). Disagreements between authors were resolved through consensus. The following data were extracted: (1) study characteristics: first author, year of publication, study design, study population (number, age, and sex), study period, study country, and institute; main outcomes (mortality, bridge to liver transplantation, length of ICU stay); (2) pre-treatment and post-treatment liver function parameters: serum bilirubin, ALT, AST, vasopressor need (mcg/kg/min), serum bile acid levels, prothrombin time, D-dimer levels; (3) changes in vital organ function: SOFA scores (Sequential Organ Failure Assessment), SAPS-II (Simplified Acute Physiology Score II), CLIF scores (Chronic Liver Failure Consortium Organ Failure), APACHE (Acute Physiology and Chronic Health Evaluation) scores; (4) safety outcomes: white and red blood cell counts, hemoglobin count, serum albumin, platelet count, neutrophil count. Only data prior to the initiation of hemoadsorption therapy and at the discontinuation of the therapy were collected.

When unavailable in writing, data estimates from visual sources were collected using software (GetData Graph Digitizer version number: v.2.26), although these estimates were not used in the meta-analysis for optimal mathematical accuracy.

### 2.5. Study Risk of Bias and Certainty of Evidence Assessment

Two authors (CT and IA) independently performed the risk of bias assessment according to the recommendations of the Cochrane Handbook [19] utilizing the Joanna-Briggs Institute’s Critical Appraisal Tool [20] for case reports and case series, ROBINS-I Risk of Bias Assessment for cohort studies [21]. Disagreements were resolved by deliberation.

The level of certainty of evidence evaluation was performed using the GRADE assessment based on the GRADE Handbook [22] and was determined using the online software GRADE Pro GDT.20 (GRADEpro Guideline Development Tool version 20, available from gradepro.org).

### 2.6. Statistical Analysis

Statistical analyses were carried out using the R statistical software (version 4.1.2.) [23]. Meta-analysis was performed for outcomes for which at least three studies reported data. The meta-analysis follows the advice of Harrer et al. [24].

For each continuous outcome, we meta-analyzed the before-treatment mean, the after-treatment mean, and their difference. We used the classical inverse variance method with the restricted maximum likelihood estimator. As only a few studies contributed to the meta-analysis, Hartung-Knapp adjustment was applied. Besides the prediction interval, heterogeneity was assessed by calculating the I^2^ measure and its confidence interval and performing the Cochrane Q test. I^2^ values of 25%, 50%, and 75% were considered low, moderate, and high heterogeneity, respectively.

In all cases, although standard deviations of the outcome before and after the treatment were available, the standard deviation of the change was missing. Following the instructions of [20], we input several different correlations from the range of −0.5 to 0.9. All the employed correlations provided more or less the same pooled results. The published results were created with an input correlation of 0.8.

Publication bias could not be assessed by visual inspection of the Funnel plot or by performing Egger’s test due to the small number of available studies.

From the meta-analyses described above, we excluded studies with one or very few observations. We visualized these excluded results on boxplots, and we tested whether the order of magnitude of the before and after values is different by performing the Wilcoxon test.

## 3. Results

### 3.1. Study Selection and Characteristics

The systematic search yielded 3022 records after duplicate removal. The selection process took place in accordance with the protocol registered on PROSPERO. The PRISMA flowchart detailing the selection process is shown in Figure 1. 

### 3.2. Main Characteristics of the Included Studies

The selection process yielded 30 eligible publications between 2011 and 2022, and a further 3 publications from the pool retrieved from the repeat systematic search. All publications reported the use of hemoadsorption for a total of 323 patients. Of those, 19 were case reports, 7 were case series (total number of patients, *n* = 84), 3 were observational studies (*n* = 130), and 1 was a registry analysis (*n* = 109). All patients presented with liver dysfunction related to acute critical illness have been treated with HA: CytoSorb (23 datasets, *n* = 232), Coupled Plasma Filtration Adsorption (4, *n* = 88), oXiris (2, *n* = 2), and CytoSorb + oXiris (1, *n* = 1). The main characteristics of the included studies along with baseline characteristics of the patients are detailed in Table 1. 

### 3.3. Primary Outcomes

The main outcomes of this study were mortality, rate of bridge to transplantation, and length of ICU stay. The lack of well-documented original research data in the literature led to none of these outcomes being able to be meta-analyzed as planned. The in-hospital mortality rate was 38% (50/130 patients) in the observational cohort studies [51,52,53]; 23% (19/82 patients) in case reports and series [27,28,29,30,31,32,33,34,35,36,37,38,39,40,41,42,43,44,45,46,47,48,49,50]; and the registry analysis by Ocskay et al. [18] reported a total of 65 cases of in-hospital mortality (59.6%): 10 at the end of HA therapy (9.2%), 60 during the ICU stay (55%), and 5 more during the out of ICU hospitalization period. Only Ocskay et al. reported the length of ICU stay (14.0 (7.0–23.0); median and IQR). No studies reported the success rate or any other descriptive outcomes in relation to bridging to liver transplantation.

### 3.4. Other Outcomes

In order to assess the use of hemoadsorption therapy in a clinical setting, we planned to review a set of exploratory outcomes. These included laboratory outcomes, safety parameters, and changes in vital organ functions. 

#### 3.4.1. Post-Treatment Organ Function Parameters

Among these outcomes, only six laboratory parameters could be meta-analyzed. Data pooled from 160 patients showed a significant reduction in total bilirubin levels post-treatment (mean difference of −4.79 mg/dL (95% CI: −6.25; −3.33), *p* = 0.002) (Figure 2). Pooled data from case series (*n* = 38) showed a non-significant reduction in serum creatinine (mean difference of −0.38 mg/dL (95% CI: −1.27; 0.5), *p* = 0.20) (Figure 3). Further analyses could only be performed using individual patient data from case reports (Figure 4). Before and after treatment values for each laboratory parameter were pooled from the case reports and summarized in box plots. Individual patient data concerning the change of these parameters are depicted by lines that connect dots that represent before and after data for each patient. These analyses showed significantly reduced AST levels (Wilcoxon *p* = 0.03) (Figure 4B) and vasopressor need (Wilcoxon *p* = 0.03) (Figure 4F) after treatment. Analyses of ALT, C-reactive protein (CRP), creatinine, and total bilirubin levels after treatment all showed non-significant tendencies for reduction (Figure 4). 

Currently, data are lacking for D-dimer, serum bile acid levels, and prothrombin time before and after treatment with hemoadsorption. Therefore, a meta-analysis could not be performed for these outcomes.

Other post-treatment organ function parameters extracted from the included articles are detailed in Appendix B Table A1. 

#### 3.4.2. Changes in Vital Organ Function

Only two studies reported SOFA score changes before and after HA therapy. Ocskay et al. [18] reported a non-significant improvement in SOFA scores of liver failure patients (mean with a CI: 0.5 (−0.3 to 1.3)), while Popescu et al. (2020) observed a significantly improved CLIF-SOFA score after HA therapy in their case series [48]. The retrospective study by Niu et al. [51] reported a significant improvement in SOFA score, but there are no data available to demonstrate this outcome. Scharf et al. [52] reported a significant improvement in SAPS-II scores after hemoadsorption therapy (mean difference of 6 ± 9, *p* = 0.01). 

Among the single-patient case reports, only Cazzato et al. [27] followed up with their patients’ SOFA scores. Their patients who underwent a hepatic resection and developed acute liver failure postoperatively improved from a SOFA score of 4 to a 2 after HA therapy. 

#### 3.4.3. Safety Outcomes

None of the included studies reported the safety outcomes planned to be presented in this review, but device-related adverse events were not reported in any of the studies.

### 3.5. Risk of Bias and Level of Evidence Certainty Assessments

The results of the risk of bias assessment and GRADE assessment of the level of evidence certainty are presented in Appendix A, respectively. 

Individual case reports were nearly free from the risk of bias according to our assessment. Case series however suffered from a lack of clearly elaborated patient enrollment strategy across the board. Overall, the risk of bias was not significant for any of the included studies.

Evidence quality is assessed to be poor by the GRADE assessment. Study designs being retrospective and observational present a major challenge in drawing reliable conclusions. Some publications on this topic might be considered “gray literature”. As such, the reliability and the quality of the evidence provided should be considered questionable. 

## 4. Discussion

ICU-acquired acute liver dysfunction in the context of a dysregulated host response and hyperinflammation is common and associated with poor short-term outcomes. Notwithstanding clinical advancements to support liver function over the last decades, diagnosis is challenging and therapeutic strategies in the form of liver support therapies are still controversially discussed, since solid data on their efficacy remain sparse. 

Therefore, we conducted this systematic review and meta-analysis on the effects of hemoadsorption on liver function in patients with confirmed liver dysfunction of various inflammatory etiologies. We found that the use of hemoadsorption for critically ill patients with acute liver dysfunction or failure seems to be safe and yields a trend towards improved liver function after hemoadsorption. 

### 4.1. Devices

There are a few different hemoadsorption technologies available on the market, of which we identified three devices that were used for ICU-acquired liver dysfunction: CytoSorb, CPFA, and oXiris. Among these, CytoSorb was by far the most frequently used. 

#### 4.1.1. CytoSorb

The CytoSorb hemoadsorber is a European CE-marked device capable to adsorb and thus remove cytokines as well as substances such as bilirubin and myoglobin from the blood compartment [54,55]. With more than 180,000 single treatments, this technology is hitherto the most frequently reported hemoadsorption device in clinical practice. 

#### 4.1.2. Coupled Plasma Filtration Adsorption

The CPFA cartridge for the removal of cytokines is a blood purification technique that separates whole blood into cellular and plasma components using a high cut-off filter. Subsequently, the plasma is filtered through an adsorbing material that can extract cytokines and then recombine the plasma and cellular components back into whole blood [56]. 

#### 4.1.3. oXiris

oXiris is a new, high-adsorption membrane filter based on the AN69 polyacrylonitrile hemofilter membrane; in addition, it undergoes additional surface treatment with polyethyleneimine (PEI) lipid A phosphate groups and heparin grafting that combines cytokine and endotoxin removal properties, renal replacement function, and anti-thrombogenic properties [57]. Surface adsorption is purely selective on endotoxin because of the specific configuration of the membrane. Conversely, bulk adsorption is nonselective and can absorb numerous mediators unselectively.

### 4.2. Outcomes

#### 4.2.1. Bilirubin

One of the most consistent findings in patients with liver dysfunction and treated with hemoadsorption is the effective reduction in bilirubin levels after hemoadsorption, which is strongly supported by the results of our current study. 

Two temporally staggered pathophysiological stages of inflammation-induced liver dysfunction can be distinguished in terms of clinical appearance and laboratory assessment. The primary dysfunction, which manifests itself within 24 h after the shock (called “ischemic hepatitis”), leads to a severe restriction of liver perfusion with centrilobular necrosis, accompanied by a massive increase in transaminases (AST, ALT) with only a slight increase in bilirubin [56]. This condition resolves within a few days after the circulation is restored. This is to be distinguished from secondary liver failure or cholestatic liver dysfunction, which is predominantly triggered by inflammatory mediators and is defined by impaired bile formation and excretion. The underlying mechanism is not an obstruction of bile ducts but a non-obstructive accumulation of bile acids and bilirubin in the liver due to a down-regulation of specific transporter molecules at the biliary side of the hepatocyte [9,58]. The mean bilirubin levels in patients included in our meta-analysis were 18.06 ± 13.26 mg/dL and 6.15 ± 2.32 mg/dL according to data from individual cases and cohorts before and after HA treatment, respectively. These levels point towards a cholestatic liver dysfunction, rather than an ischemic type.

There is some evidence from experimental studies that high bilirubin concentrations inhibit the non-specific defense mechanisms of neutrophil granulocytes. Because of the antioxidant properties of bilirubin, the bactericidal effect of reactive oxygen species can be inhibited, which enhances the systemic spread of bacteria in an already critical phase [59].

#### 4.2.2. ALT, AST, Bile Acid, Ammonia

However, hemoadsorption may effectively remove not only bilirubin from the blood but also, as shown in two recent in vitro experiments, effectively remove bile acids [60,61]. These results indicate that hemoadsorbents may remove hydrophobic, albumin-bound bile acids better than CRRT filters. Although aminotransferases, levels of bile acid, and serum ammonia are regularly used in clinical practice as markers for liver function, there is hardly any clinical evidence on the effect of hemoadsorption on these parameters. In fact, a recent study found that ammonia elimination is mainly achieved by the dialysis filter rather than CytoSorb [62]. Furthermore, Scharf et al. hypothesized that the molecular weight of AST, ALT, and GGT makes the transaminase reduction unlikely, and the significant reduction observed suggests a potential improvement in liver function [52]. Therefore, the direct removal of substances versus secondary effects during hemoadsorption therapy remains an unresolved issue. Future studies are needed, in which concentrations of the substances of interest should be measured in the in-flow line (pre-adsorber) and the in the out-flow line (post-adsorber) to determine the clearance of these molecules by the hemoadsorber.

#### 4.2.3. Clinical Outcomes and Safety

This review establishes that there is a critical lack of hard evidence on clinical outcomes associated with hemoadsorption therapy. Although the device itself does not seem to have any adverse effects or complications associated with its use, there is no systematically generated evidence for this claim to be sufficiently reliable. The existing evidence on clinical outcomes is either deemed to be of low quality according to the GRADE assessment or needs to be corroborated and complemented by more studies. The registry analysis from 2019 includes assessments by involved clinicians on whether hemoadsorption therapy improved, deteriorated, or did not affect the clinical status of the patients. While clinicians assessed 68.9% (*n* = 75) of patients’ conditions to have been improved by the therapy, 15.6% (*n* = 17) of patients did not show any change and 4.8% (*n* = 5) deteriorated. Due to the lack of comparative studies, it is impossible to draw solid conclusions for such an outcome. However, the current lack of evidence should not be misconstrued as a lack of interest in the topic nor as a demonstration of the inefficacy of the therapy.

### 4.3. Implications for Research and Practice

Two recent meta-analyses on different extracorporeal liver-support devices showed that this issue is still unsolved, and the level of evidence is so low that recommendations on which approach is the best cannot be made [1,63]. Hemoadsorption is relatively simple to apply, and according to some recent data, it may even be superior to Molecular Adsorbent Recirculating System (MARS). In a recent in vitro study, CytoSorb was found superior to MARS as far as bilirubin, bile acid, ammonia, and cytokine removal are concerned [57]. However, large prospective data or results of randomized trials are still missing. Furthermore, it would also be important to consider alternative study endpoints, such as the change in levels of mercaptans, idols, tryptophane, and albumin binding capacity [64]. Such studies in the future could fill in the gaps in the currently available evidence and knowledge on HA therapy, particularly those associated with clinical outcomes for patients with acute liver dysfunction.

This study has been conducted in the framework of Academia Europaea’s position on the cycle model of translational medicine for community healthcare benefit [65,66]. Accordingly, our findings and elaboration are aimed towards summarizing and contextualizing discussions around this highly important subject to generate new hypotheses and guide further research.

### 4.4. Limitations

The current study has several limitations. First and foremost, the limitation is imposed by the lack of randomized controlled clinical trials in the literature. Second, several of the included studies are case reports and series, which limit the generalizability of the findings from the meta-analyses. Third, several included studies fail to report the sex and ethnicity of the patients, which are both important factors to consider in the clinical overview.

## 5. Conclusions

The current systematic review and meta-analysis provide further support that adjuvant therapy with hemoadsorption is a feasible, safe, and effective method to reduce circulating bilirubin levels and may have direct and/or indirect effects on other liver-related potentially toxic metabolites. However, the quality of evidence is still low and very little is known about the clinical effects of the therapy. Therefore, our results highlight the need for adequately designed clinical trials with the above-mentioned parameters as the main outcomes.

## Figures and Tables

**Figure 1 biomedicines-12-00067-f001:**
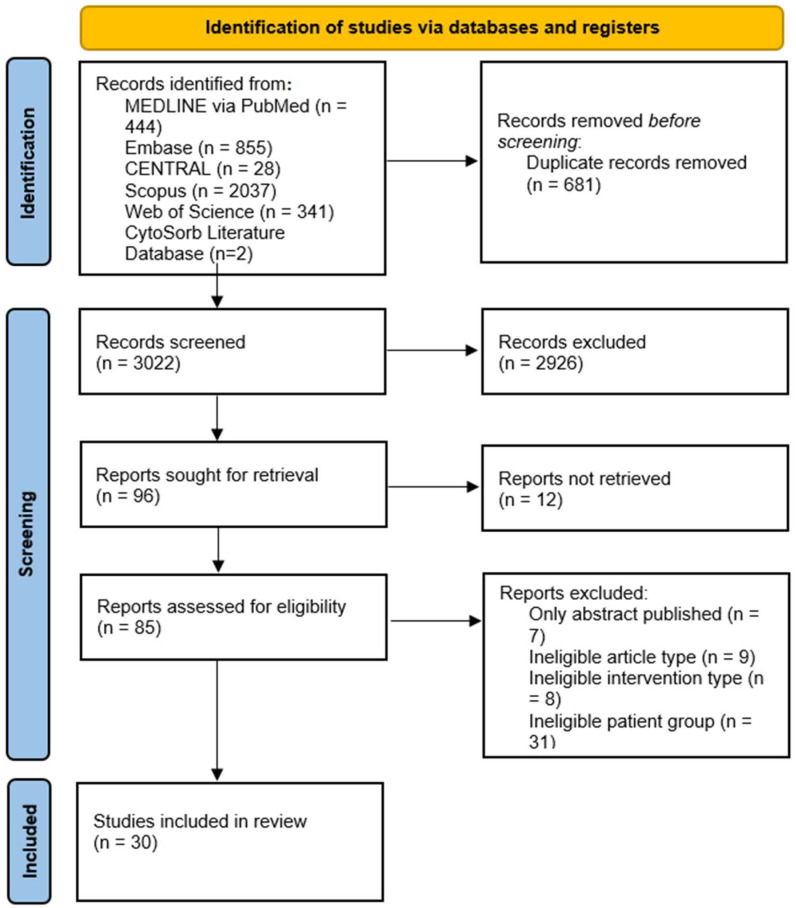
PRISMA flowchart of included studies.

**Figure 2 biomedicines-12-00067-f002:**
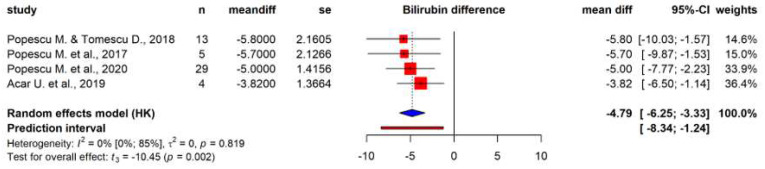
Total bilirubin levels. Forest plot of total bilirubin levels pre- and post-treatment with hemoadsorption [46,47,48,50].

**Figure 3 biomedicines-12-00067-f003:**
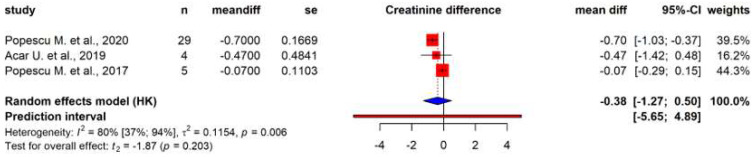
Creatinine levels. Forest plot of serum creatinine levels pre- and post-treatment with hemoadsorption [47,48,50].

**Figure 4 biomedicines-12-00067-f004:**
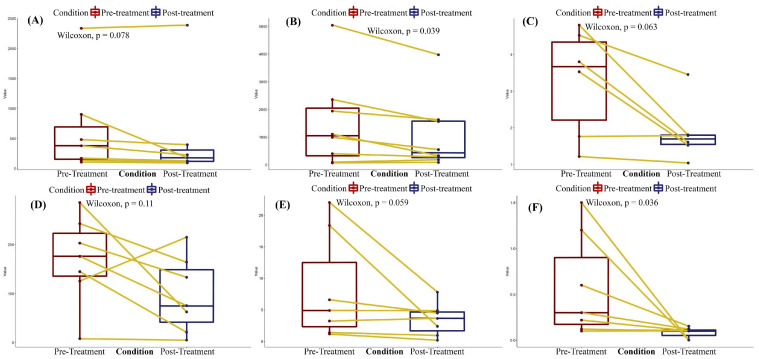
Box plots of individual case data: (**A**) alanine aminotransferase (ALT), (**B**) aspartate aminotransferase (AST), (**C**) bilirubin, (**D**) creatinine, (**E**) C-reactive protein (CRP), and (**F**) vasopressor need. Data were pooled from individual case reports and presented as box plots, representing pre- and post-treatment values. Changes in these parameters for each case are also depicted by lines connecting pre- and post-treatment values.

**Table 1 biomedicines-12-00067-t001:** Study and baseline characteristics of included studies.

Publication Data		Study Design	Number of Patients	Age	Used Device	Intervention	Number of Sessions
First Author	Year of Publication						
Gunasekera, A.M. [25]	2022	Case report	1	54 ^a^	CytoSorb	CRRT with CytoSorb	1
Ruiz-Rodriguez, J.C. [26]	2022	Case report	1	50 ^a^	CytoSorb	CVVHDF with CytoSorb	1
Cazzato, M.T. [27]	2019	Case report	1	No data	CytoSorb	CRRT with CytoSorb (24 h)	4
Daza, J.L. [28]	2022	Case report	1	41 ^a^	CytoSorb	SLED combined with CytoSorb (12 h)	2
Hinz, B. [29]	2015	Case report	1	72 ^a^	CytoSorb	CVVHD with CytoSorb (24-6-24 h)	3
Köhler, T. [30]	2021	Case report	1	29 ^a^	CytoSorb	CRRT with CytoSorb (24 h)	Unclear
Lau, C.W.M. [31]	2021	Case report	1	47 ^a^	oXiris	Blood purification with oXiris (5 days in total)	No data
Li, Y. [32]	2020	Case report	1	35 ^a^	oXiris	CVVH with oXiris (24 h)	2
Manohar, V. [33]	2017	Case report	1	22 ^a^	CytoSorb	Extracorporeal cytokine hemofiltration (12 h)	1
Markovic, M. [34]	2020	Case report	1	31 ^a^	CytoSorb and oXiris	CytoSorb (day 1) and oXiris (day 2)	2
Moretti, R. [35]	2011	Case report	1	27 ^a^	CPFA	CPFA (24 h)	5
Piwowarczyk, P. [36]	2019	Case report	1	57 ^a^	CytoSorb	CytoSorb with anticoagulated CVVHD (24 h)	2
Tomescu, D. [37]	2018	Case report	1	17 ^a^	CytoSorb	CytoSorb (before and throughout liver transplantation)	1
Wiegele, M. [38]	2015	Case report	1	44 ^a^	CytoSorb	CytoSorb (6 h)	2
Lévai, T. [39]	2019	Case report	1	42 ^a^	CytoSorb	CytoSorb with anticoagulated CVVRRT	4
Manini, E. [40]	2019	Case report	1	62 ^a^	CytoSorb	CytoSorb with anticoagulated CVVRRT	1
Popescu, M. [41]	2017	Case report	1	47 ^a^	CytoSorb	CytoSorb (24 h)	4
Kogelman, K. [42]	2021	Case report	1	45 ^a^	CytoSorb	CytoSorb with CRRT (in CVVHD mode)	3
Breitkopf, R. [43]	2020	Case report	1	40 ^a^	CytoSorb	CytoSorb with CRRT (in CVVHD mode)	2
Ullo, I. [44]	2017	Case series	9	21–63 ^b^	CPFA	CPFA with citrate anticoagulation	No data
Popescu, M. [45]	2017	Case series	5	49 ± 13 ^c^	CytoSorb	CytoSorb with CVVHF	No data
Popescu, M. and Tomescu, D. [46]	2018	Case series	13	46 ± 17 ^c^	CytoSorb	CytoSorb with CVVHF	No data
Maggi, U. [47]	2013	Case series	2	22–64 ^b^	CPFA	CPFA	3
Popescu, M. [48]	2020	Case series	29	34 ± 14 ^c^	CytoSorb	CytoSorb with CVVHDF	3
Dhokia, V.D. [49]	2019	Case series	3	51–71 ^b^	CytoSorb	CytoSorb with CVVHDF (1); CytoSorb with Prismaflex (1); CytoSorb with CRRT (1)	2
Acar, U. [50]	2019	Case series	4	26–73 ^b^	CytoSorb	CytoSorb with CVVHD	No data
Ocskay, K. [18]	2021	Registry analysis	109	49.2 ± 17.1 ^c^	CytoSorb	Varies: CytoSorb alone or CytoSorb with CRRT	2
Niu, D.G. [51]	2019	Retrospective observational study	76	51.4 ± 15.6 ^c^	CPFA	CPFA with CRRT	No data
Scharf, C. [52]	2021	Retrospective observational study	33	55 (18–76) ^d^	CytoSorb	CytoSorb	1
Praxenthaler, J. [53]	2022	Retrospective observational study	21	74 (58–80) ^d^	CytoSorb	CVVHD with CytoSorb	varies

^a^ Individual data, ^b^ range (min–max), ^c^ mean ± standard deviation, ^d^ median (minimum range–maximum range).

## Data Availability

All data used for the meta-analysis originate from publicly available publications.

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
