# Peer review of "Hemoadsorption Therapy for Critically Ill Patients with Acute Liver Dysfunction: A Meta-Analysis and Systematic Review"

_biomedicines, 2023, doi:10.3390/biomedicines12010067_

Round 1
Reviewer 1 Report
Comments and Suggestions for Authors
The paper is interesting and has a high scientific impact, but it should be reworked.
The abstract needs to be rewritten as it repeats the material and methods section and does not present the point of the review
Additionally, the introduction requires revision. The article numbers should be placed before the period rather than after it. The paragraph from lines 62 to 74 is quite complex to understand, so it would be better to paraphrase it.
The references section does not adhere to the requirements of the journal.
The aim is highlighted and well described.
The section Material and Methods is well described, and all methods are easily repeated.
Figure 4 is complicated to understand. Present the information more appropriately.
The discussion section should be expanded and supported with more clearly stated statements. Overall, the article is a meta-review and needs to be expanded.
I recommend a major revision оf the manucript.
Comments on the Quality of English LanguageThe English language needs to be edited.
Reviewer 2 Report
Comments and Suggestions for Authors
The manuscript titled “Hemoadsorption Therapy for Critically Patients with Acute Liver Dysfunction: A Meta-Analysis and Systematic Review” by Turan, C.; et al. is a Review work where the authors summarize the most relevant factors to control (bilirubin levels, prothrombin time, alanine aminotransferase and aspartate aminotransferase levels, among others) during hemoadsorption therapies for those patients affected by liver diseases. The meta-analysis study provided the varience of the tested leves before and after the hemoadsorption treatment. The methodology used in this study could be extandable for other type of human malignancies.
The scientific content is interesting and the sections are well-designed. However, it exists some points that need to be addressed (please, see them below detailed point-by-point). The most relevant outcomes found by the authors can contribute to in the growth of many fields like the clinical&healthcare overall related to the treatment of liver diseases. This knowledge could aid in the design of the next-generation of therapies against liver disorders by selecting the pivotal parameters to be monitored during the treatment. For this reason, I will recommend the present scientific manuscript for further publication in Biomedicines once all the below described suggestions will be properly fixed.
Here, there exists some points that must be covered in order to improve the scientific quality of the manuscript paper:
1) ABSTRACT. “The search yielded (…) the use of HA for (…)” (lines 21-22). Please, the authors should define the full-name once one term appears in the text for the first time. This point should be taken into account for the rest of the main manuscript body text (e.g. “ALT” and “CRP” in line 26, “ICU” in line 33, among many other cases).
2) KEYWORDS (OPTIONAL). The authors should consider to add the term “database meta-analysis” in the keyword list.
3) INTRODUCTION. “When advancing into more sever states (…) hepatic encephalopathy or brain dysfunction (…) liver failure” (lines 41-43). Even if I agree with this statement provided by the authors, a relevant reference of this field is missing [1].
[1] Lu, K. Cellular Pathogenesis of Hepatic Encephalopathy: An Update. Biomolecules 2023, 13, 396. https://doi.org/10.3390/biom13020396.
4) Linked to the above described point, it is also neccesary to point out how the production of redox species or positive divalent cations can negatively affect at cellular [2] or biomolecular [3] levels and they could trigger the progression of human hepatic diseases.
[2] Allameh, A.; Niayesh-Mehr, R.; Aliarab, A.; Sebastiani, G.; Pantopoulos, K. Oxidative Stress in Liver Pathophysiology and Disease. Antioxidants 2023, 12, 1653. https://doi.org/10.3390/antiox12091653.
[3] Vega, S.; Neira, J.L.; Marcuello C.; Lostao, A.; Abian, O.; Velazquez-Campoy, A. NS3 protease from hepatitis C virus: biophysical studies on an intrinsically disordered protein domain. Int. J. Mol. Sci. 2013, 14, 13282-13306. https://doi.org/10.3390/ijms140713282.
5) “However, despite good correlations between bilirrubin plasma concentrations and mortality in several critically ill conditions” (lines 50-51). Could the authors furnish some quantitative information in this regard?
6) MATERIALS AND METHODS. This section perfectly explains all the methodology employed to devote this research. No actions are requested from the authors.
7) RESULTS. Figure 1 (line 174). Please, the authors should erase the underline in red and blue colours of the terms “Embase”, “CytoSorb” and “removed”, respectively.
8) Table 1 (line 187). The sex ratio (male/female) should be added since this parameter is a key factor to consider. Furthermore, the race should be also taken into account at least explained in the main manuscript body text.
9) Figure 4 (line 225). Could the authors increase the line thickness of the box-plots? This could significantly aid to the readers to better visualize the displayed content of this Figure.
10) DISCUSSION. “4.1.1. CytoSorb (…) in clinical practice” (lines 273-277). What is the effect of CytoSorb treatment on the progress of patients with hepatic diseases?.
11) CONCLUSIONS. The authors clearly remarked the most relevant outcomes found in this Review work and also potential future lines to pursue this research: “Therefore, our results highlight the need for adequately designed clinical trials (…) outcomes” (lines 369-371). No actions are requested from the authors concerning this section.
12) REFERENCES. The references are mostly in the proper format style of Biomedicines. Nevertheless, the authors should take a look to homogenize some final aspects susceptible to be improved (e.g. The publication year should be highlighted in bold and some references match with this criteria but not in other cases).
Comments on the Quality of English Language
The authors should take a last look to polish final editing details and fix those typos susceptible to be improved.
Round 2
Reviewer 1 Report
Comments and Suggestions for Authors
Accept in present form